# Modeling of Human Rabies Cases in Brazil in Different Future Global Warming Scenarios

**DOI:** 10.3390/ijerph21020212

**Published:** 2024-02-11

**Authors:** Jessica Milena Moura Neves, Vinicius Silva Belo, Cristina Maria Souza Catita, Beatriz Fátima Alves de Oliveira, Marco Aurelio Pereira Horta

**Affiliations:** 1Biosafety Level 3 Facility, Oswaldo Cruz Institute, Fiocruz, Rio de Janeiro 21040-900, Brazil; marco.horta@fiocruz.br; 2Laboratory of Parasitology, Federal University of São João del-Rei, Divinopolis 36307-352, Brazil; viniciusbelo@ufsj.edu.br; 3Department of Geographic Engineering, Geophysics and Energy, University of Lisbon, Lisbon 1649-004, Portugal; cmcatita@fc.ul.pt; 4Oswaldo Cruz Institute, Fiocruz, National School of Public Health and Regional Office of Piaui, Teresina 64001-350, Brazil; beatriz.oliveira@fiocruz.br

**Keywords:** human rabies, climate change, distribution modeling, CMIP6, Brazil

## Abstract

Bat species have been observed to have the potential to expand their distribution in response to climate change, thereby influencing shifts in the spatial distribution and population dynamics of human rabies cases. In this study, we applied an ensemble niche modeling approach to project climatic suitability under different future global warming scenarios for human rabies cases in Brazil, and assessed the impact on the probability of emergence of new cases. We obtained notification records of human rabies cases in all Brazilian cities from January 2001 to August 2023, as reported by the State and Municipal Health Departments. The current and future climate data were sourced from a digital repository on the WorldClim website. The future bioclimatic variables provided were downscaled climate projections from CMIP6 (a global model ensemble) and extracted from the regionalized climate model HadGEM3-GC31-LL for three future socioeconomic scenarios over four periods (2021–2100). Seven statistical algorithms (MAXENT, MARS, RF, FDA, CTA, GAM, and GLM) were selected for modeling human rabies. Temperature seasonality was the bioclimatic variable with the highest relative contribution to both current and future consensus models. Future scenario modeling for human rabies indicated a trend of changes in the areas of occurrence, maintaining the current pace of global warming, population growth, socioeconomic instability, and the loss of natural areas. In Brazil, there are areas with a higher likelihood of climatic factors contributing to the emergence of cases. When assessing future scenarios, a change in the local climatic suitability is observed that may lead to a reduction or increase in cases, depending on the region.

## 1. Introduction

Human rabies is a neglected disease that has a profound impact on public health in impoverished communities and low- to middle-income countries [1]. With the global fatality rate of 98.4%, rabies has an annual incidence of approximately 60,000 cases spanning over 150 countries. It is characterized by an acute ailment affecting the central nervous system, clinically manifesting as progressive encephalitis [2]. The virus is primarily transmitted through the saliva and secretions of infected animals, with susceptibility across all mammalian species, thus contributing to disease dissemination [3]. Globally, rabies is responsible for more than 59,000 annual fatalities, with African and Asian regions collectively accounting for over 95% of the confirmed human cases [4,5]. The mass vaccination of domestic animals has proven to be a highly effective strategy for reducing the disease prevalence of rabies in various regions, including Africa, Asia, Europe, and the Americas [6]. Furthermore, more than 15 million individuals worldwide receive pre-exposure prophylaxis annually, and over 29 million individuals receive post-exposure vaccination, thereby contributing to disease prevention and reducing mortality on a global scale [2].

In Brazil, the National Rabies Prophylaxis Program (PNPR) has instituted four primary surveillance strategies, which include an annual national vaccination campaign for dogs and cats, human rabies prophylaxis, case notification and investigation, and the monitoring of viral circulation [7]. With the implementation of these strategies over the past decades, the country has achieved significant results in the control of urban rabies, marked by a sharp decline in the number of cases, with only sporadic and accidental occurrences prevailing [8]. Through its specific variant, dogs were the focus of public health in Brazil as the primary rabies transmitter to humans in urban areas [9]. However, since 2016, there has been an increase in cases attributed to bats, and rabies cases in dogs and cats have been identified with variants of wildlife [7]. In 2017, among the six detected rabies cases, none were related to dog bites, five cases were attributed to bats, and one case was associated with a feline bite. In 2018, all 11 cases described were attributed to bats; additionally, in 2019, the single reported case was caused by a feline bite from an animal infected with the AgV-3 bat antigenic variant. In 2020, the Ministry of Health reported only one case of bat-transmitted rabies. In 2021, a case of human rabies was recorded, linked to a variant of a wild canid (*Cerdocyon thous*); moreover, in 2022, the five confirmed rabies cases were associated with the antigenic variant of bat AgV-3 [8].

These observations complicate rabies control and support a shift in public health focus, as bat population control measures are not targeted in Brazil [10]. Bats have a significant epidemiological importance in the sylvatic aerial cycle. Among the 182 officially registered species in Brazil, the rabies virus has been isolated in 31 species, with the hematophagous bat species *Desmondus rotundus* (*D. Rotundus*) being the primary vector for viral transmission in rural areas, especially among production animals, such as cattle and horses [11,12]. Factors such as habitat alterations due to deforestation and the increasing availability of livestock for hematophagous species, driven by the expansion of cattle farming as a food source in Brazil, may have favored bat-transmitted rabies in recent years, resulting in a shift in the epidemiological profile [13,14]. It is estimated that in the entire Latin American region, rabies in herbivores causes annual losses of hundreds of millions of dollars due to the deaths of thousands of cattle, in addition to indirect expenditures on bovine vaccination and post-exposure treatments (post-bite vaccination) for individuals who have had contact with suspected animals [15].

Moreover, the current rabies scenario in Brazil is marked by concerns about the potential mass reintroduction of urban rabies through variants from sylvatic reservoirs, with the potential to reverse the epidemiological profile [16,17]. Bats play an increasingly significant role in urban environments, with periodic colony displacements becoming more pronounced [18,19]. This is because of the deforestation associated with urban expansion, resulting in the loss of natural habitats of bats, such as forests and riparian areas, and reducing the availability of shelter, breeding, and feeding sites for these species [20,21]. These alterations directly impact the dynamics of bat communities in various regions of Brazil, leading cities to become favorable locations by offering better shelter conditions (pipes under highways, sewers, and building crevices) and food, resulting in an increased proximity to humans and elevated susceptibility to rabies [22,23].

Climate change also influences the ecology and population dynamics of bats and has significant implications for rabies transmission [24]. Environmental temperature variation, for example, has both direct and indirect effects on hematophagous bats, impacting their activities, feeding patterns, reproduction, and migration [25,26,27]. Studies have indicated that extreme temperatures can affect thermoregulation in these species, triggering metabolic changes. Excessively low temperatures tend to reduce bat activity, while excessively high temperatures may force them to remain inactive during daylight hours [28,29]. Climate-related variations in temperature and precipitation also influence the geographic distribution of hematophagous bats and availability of food sources [30]. In regions where temperature and precipitation patterns undergo changes that affect the migration of prey, such as wildlife, birds, and livestock, bats may adjust their movement patterns to seek food elsewhere [31]. As climatic conditions undergo transformation, the areas inhabited and frequented by these bats may shift, prompting the search for new territories that offer more suitable conditions for their survival [32]. This, in turn, can result in an increased proximity to human communities and other animals, potentially heightening interactions and the associated risks of coexistence.

This array of transformations in the habitats of hematophagous bats, including deforestation, the conversion of natural ecosystems into pastures for intensive livestock farming, and local and regional climate variations resulting from global warming, has the potential to alter the range of bat species, with implications for the occurrence of rabies cases in previously unaffected regions [33,34]. The studies investigating the spatiotemporal distribution of the rabies virus in bats and other mammalian species have emphasized the influence of factors such as temperature, precipitation, and the El Niño–Southern Oscillation phenomenon on the emergence of rabies outbreaks in specific locations during different seasons of the year [28,30]. Furthermore, studies have highlighted environmental conditions, particularly temperature variations, when analyzing the effects of climate change on rabies virus transmission, especially in livestock, such as cattle [35,36].

The application of species distribution modeling to assess the extent of dispersal and the impact of climate change on future suitability plays a pivotal role in the development of appropriate management strategies for the conservation and sustainability of species habitats [37]. By projecting future scenarios for hematophagous bats in the border region between the USA and Mexico, researchers identified areas in South Texas that could become suitable for the occurrence of these species by 2070 [38]. Additionally, in a study utilizing a predictive analysis based on future climate scenarios for *D. rotundus*, the authors identified highly favorable habitats for the species across Mexico and Central America, demonstrating that temperature and precipitation variations could explain the expansion of these species in these regions [29]. However, owing to the associated uncertainty, the climate models have not reached unanimous conclusions regarding future scenarios.

Rabies poses a significant risk to public health and should therefore be included in the scope of health surveillance services. This is particularly relevant in the context of research on the monitoring and prediction of rabies cases. Such studies should not only consider socioeconomic and cultural factors, but also climatic parameters that affect animals serving as disease reservoirs in a daily manner. Therefore, our study aimed to analyze the spatial distribution of human rabies cases in Brazil, considering different global warming scenarios to assess their impact on the likelihood of new cases emerging.

## 2. Material and Methods

### 2.1. Health Data

We obtained the records of reported cases of human rabies in all Brazilian cities from January 2001 to August 2023 provided by the State and Municipal Health Departments through the Information System for Notifiable Diseases [39]. The collected data were subjected to georeferencing, resulting in the creation of central geographic coordinates (centroids) for each municipality with the reported cases. All data were analyzed using an Excel spreadsheet. Brazil, situated in South America, ranks as the fifth largest country in the world in terms of the territorial area and the sixth largest with respect to its population. Its remarkable geographic and climatic diversity spans a vast range of ecosystems from the Amazon Rainforest to the Pantanal and the Cerrado (Appendix A Appendix A). The Brazilian territory is subdivided into five regions: the northern, northeastern, midwestern, southeastern, and southern regions. Each region has distinct characteristics influenced by factors such as the climate, topography, and natural resources (Figure 1) [40,41].

### 2.2. Climate Data

The meteorological data, including temperature, relative humidity, and precipitation records, were acquired from a digital repository available on the WorldClim website [42]. Additionally, these data include bioclimatic variables calculated based on the monthly time series of temperature and precipitation. These bioclimatic variables encompass the annual and monthly patterns, seasonality, and exceptional or limiting climatic elements, such as extreme temperatures observed during the warmest or coldest months [43]. These variables have significant relevance in the context of species distribution modeling, ecological niche investigations, and research dedicated to climate change dynamics.

The WorldClim climatic variables include a series of parameters, such as the annual mean temperature (BIO1), mean diurnal temperature range (BIO2), isothermality (BIO3), temperature seasonality (BIO4—standard deviation100), maximum temperature of the warmest month (BIO5), minimum temperature of the coldest month (BIO6), the annual temperature range (BIO7), the mean temperature of the wettest quarter (BIO8), the mean temperature of the driest quarter (BIO9), the mean temperature of the warmest quarter (BIO10), the mean temperature of the coldest quarter (BIO11), annual precipitation (BIO12), precipitation of the wettest month (BIO13), precipitation of the driest month (BIO14), precipitation seasonality (BIO15—coefficient of variation), precipitation of the wettest quarter (BIO16), precipitation of the driest quarter (BIO17), precipitation of the warmest quarter (BIO18), and precipitation of the coldest quarter (BIO19).

For the analysis of the current climatic conditions, georeferenced files containing 19 bioclimatic variables corresponding to the period 1970–2000 were acquired at four spatial resolutions—30 s, 2.5 min, 5 min, and 10 min—with area sizes ranging from 0.86 km^2^ to 344 km^2^ at the equator. The selected resolution for the model was 2.5 min (0.86 km^2^). Each download comprises a compressed file in “zip” format containing 12 GeoTiff files, one for each month of the year (from January to December), with image dimensions of 2160 × 1080 pixels and a resolution of 96 dpi.

In the context of future climatic condition analysis, bioclimatic variable data were generated through future climate projections at the same 2.5 min resolution (0.86 km^2^). These projections were made available through the digital repository on the WorldClim website [40], downscaled from CMIP6 (a global model ensemble used in the Intergovernmental Panel on Climate Change (IPCC) climate change analyses), and relied on the regionalized climate model HadGEM3-GC31-LL. This model represents a specific variant of a global climate model developed by the Met Office Hadley Center in the United Kingdom. The HadGEM3 (Hadley Centre Global Environment Model, version 3) is part of a family of climate models used to simulate atmospheric and oceanic conditions of the Earth on a global scale. The designation “regionalized” indicates that it can be configured to simulate climate conditions at more detailed regional scales than conventional global models. These models play crucial roles in understanding climate change, forecasting the future climate, and assessing the potential impacts of different greenhouse gas (GHG) emission scenarios [44].

Three future socioeconomic scenarios were considered, which are components of a set of shared socioeconomic pathways (SSP): CO_2_ emissions (SSP1-2.6, SSP2-4.5, and SSP5-8.5) for four distinct periods spanning 2021–2100 (2021–2040, 2041–2060, 2061–2080, and 2081–2100). These scenarios were designed to explore various potential trajectories of global socioeconomic development considering the different levels of GHG emissions over time. Their construction is based on the assumptions regarding the demographic, technological, economic, and political changes; additionally, their utility lies in exploring distinct paths for global development. These scenarios enable climate models to project the evolution of climate change in diverse socioeconomic contexts [45].

The SSP1-2.6 scenario, titled “Sustainability”, outlines a sustainable future characterized by low GHG emissions. The SSP2-4.5 scenario, labeled “Middle of the Road”, represents a trajectory of moderate emissions, reflecting a world in which measures for emissions reduction are implemented moderately without extreme policy or technological changes. However, the SSP5-8.5 scenario, titled “Fossil-fueled Development”, portrays a future in which GHG emissions continue to increase due to a persistent dependence on fossil fuels [46].

### 2.3. Data Analysis

To generate current and future climate suitability maps for rabies, we applied a niche modeling approach employing seven distinct algorithms. This approach was implemented using the biomod2 package in R “https://www.R-project.org/ (accessed on 18 October 2022)”. The algorithms used included Generalized Linear Regression (GLM), Generalized Additive Model (GAM), Classification Tree (CTA), Flexible Discriminant Analysis (FDA), Multivariate Adaptive Regression Splines (MARS), Random Forest (RF), and Maximum Entropy (MAXENT). These algorithms are based on the presence/absence of data, estimations of environmental similarities, and the identification of the points of intersection between known locations of species occurrence and regions that remain unknown. Thus, areas showing greater similarity to the locations where the species has been recorded are considered regions with a high suitability probability [47].

During the modeling process, human rabies cases were considered as binomial outcomes (dependent variables), while bioclimatic variables were used as explanatory variables to fit the models. For this purpose, we selected variables with the highest percentage contributions for the seven predictive algorithms (GLM, GAM, CTA, FDA, MARS, RF, and MAXENT) in both current and future scenarios. We adopted the default settings for all algorithms, considering 10,000 pseudo-absences as the background data for each algorithm. The models were developed using training sets, and those that demonstrated the best performance were selected. After the selection of the bioclimatic variables that best matched the model performance, we evaluated the Area under the Receiver Operating Characteristic Curve (AUC-ROC). The AUC values range from 0 to 1, with ranges of 0.5 to 0.7 indicating low model performance, 0.7 to 0.9 suggesting an acceptable model, and values above 0.9 indicating excellent model performance. For the final model generation, we retained those with ROC scores ≥ 0.9.

We mapped the binary distributions (0 and 1) of the raw dependent variables. After the modeling process, we produced consensus maps with ensemble models in which we assessed the climate suitability using an index ranging from 0 to 1. In this index, 0 represents areas with a low predicted probability of adaptation to climate conditions (indicated in blue), whereas 1 represents regions with a high probability of suitable climate conditions (indicated in red) for human rabies cases. We developed a final ensemble model for each combination of climate scenarios and periods using the average of the individual ensemble models. As a result, we obtained one ensemble prediction for current climate suitability and four ensemble predictions for the period from 2021 to 2100 (2021–2040, 2041–2060, 2061–2080, and 2081–2100), reflecting three future socioeconomic scenarios related to CO_2_ emissions (SSP1-2.6, SSP2-4.5, and SSP5-8.5). The selected scenarios for the climate model are integrated into the CMIP6 (Coupled Model Intercomparison Project Phase 6), which encompasses a series of socioeconomic scenarios known as SSPs (Shared Socioeconomic Pathways). These SSPs are categorized according to levels of greenhouse gas emissions and are developed to represent various potential trajectories of socioeconomic development. When used in conjunction with climate models, it becomes possible to assess the impacts of climate change under a variety of conditions [48].

To gauge the magnitude of regional modifications leading to climatic suitability in future scenario models, we calculated the percentage variation using the following formula: [(Future Analysis − Current Analysis)/Current Analysis × 100]. This method quantifies the proportion of model projections that categorize a specific cell on a map as an area of climatic suitability, elucidates regions that have undergone changes over the years, and aids in the understanding of potential patterns and trends. To more clearly visualize the regions that exhibited a greater inclination towards climatic suitability, a grayscale representation was adopted. White indicates the areas where no changes contributing to climatic suitability were observed, whereas black indicates transformations that resulted in an increase in suitability under future climatic conditions. For these analyses, we used R Studio software version 1.4, with the biomod2, raster, rgdal, and ncdf4 packages [49]. Maps were created using QGIS software version 3.28.

## 3. Results

The variables that stood out in the consensus model, revealing climate suitability for the occurrence of human rabies in Brazil, included temperature seasonality (BIO4), minimum temperature of the coldest month (BIO6), and precipitation seasonality (BIO15). Among these variables, BIO4 showed the most significant relative contribution to the consensus models, as shown in Table 1.

The consensus models developed using human rabies data to assess the climate suitability in current and future scenarios averaged an Area Under the Curve (ROC) value of 0.90, with a sensitivity of 87% and specificity of 81%. These results demonstrate the robustness of the model in predicting climate suitability. Under the current climatic conditions, the model highlighted areas represented in red on the map, indicating locations with high climatic suitability. This implies a higher likelihood that climatic factors contribute to the emergence of new rabies cases in humans. These areas of higher probability were primarily concentrated in the northern and northeastern regions, as shown in Figure 2.

The consensus models applied to three future socioeconomic scenarios (SSP1-2.6, SSP2-4.5, and SSP5-8.5) revealed a trend of modification in the areas of occurrence, while maintaining the current pace of global warming, population growth, socioeconomic instability, and the loss of natural areas. It was observed that some current areas, such as the northern and northeastern regions, remained continuously suitable under future climate conditions, both in the optimistic (SSP1-2.6) scenario of CO_2_ emissions and in the pessimistic (SSP5-8.5) scenario, characterized by a significant increase in CO_2_ emissions over time (Figure 3).

In the analysis of the percentage change in the consensus models for future scenarios, it was possible to visualize the percentage of local suitability change in all scenarios and periods using grayscale. By 2060, certain regions (northern, northeastern, and southeastern regions) maintained their suitability in all scenarios; the southern region experienced a decrease in suitability, and the central–western region showed increased climate suitability, both in the optimistic (SSP1-2.6) and the most pessimistic (SSP5-8.5) scenarios regarding CO_2_ emissions over time. Furthermore, disparities in the patterns of climate suitability intensity were observed in the southern region of the national territory during the time interval between 2041 and 2060, as outlined in the scenarios under consideration. In both analyzed scenarios (SSP1-2.6 and SSP2-4.5), which were considered optimistic and intermediate, respectively, regarding greenhouse gas emissions (GHG), there was an increase in the intensity of climate suitability. Conversely, in the pessimistic scenario (SSP5-8.5), there was a notable reduction in intensity over the same period. From 2081 onwards, especially in the most pessimistic scenario of gas emissions (SSP5-8.5), a substantial increase in the climate suitability was noted in the northeastern, central–western, southern, and southeastern regions. By analyzing the local changes in climate suitability in future scenarios, a trend of modification in the areas of occurrence and an increase in the likelihood of new cases of rabies in humans were observed (Figure 4).

## 4. Discussion

The use of species distribution modeling to determine the extent of dispersion and the impact of future climate changes is a crucial component when developing appropriate management practices aimed at the conservation and sustainability of species habitats in the future [37,49]. Over the past few decades, the Earth’s increasing global temperature has led to significant changes in ecological niches, resulting in the expansion and contraction of these niches to which animal species have had to adapt [50,51]. Various niche modeling studies have focused on the analysis of animal species to map their current habitats and predict the impact of climate change on the suitability of these habitats in the future [29,52,53,54]. In this study, a pioneering effort was made to map the occurrence locations using records of reported human rabies cases under different global warming scenarios. The goal was to identify the areas most potentially suitable for the emergence of new cases of this disease. Our results demonstrated robustness in predicting the climatic suitability, as they exhibited ROC values of 0.90, and illustrated how the combination of disease distribution models and the use of ensemble mapping can constitute useful tools for developing testable hypotheses related to disease distribution and potential future dissemination patterns, considering the impact of future climatic conditions.

The modeling approach employed in this study provides predictions that resemble the analyses conducted on the distribution of vampire bats, *D. Rotundus*, under the current and future climatic scenarios [28,29,30]. For instance, the bioclimatic variables that stood out in the consensus model of this study paralleled the findings of other researchers and aligned with the ecological requirements of *D. rotundus*, which needs to inhabit warmer regions and avoid areas with severe winters [55,56]. These variables included temperature seasonality (BIO4) and the mean temperature of the coldest month (BIO6). The data from this study agree with those of previous studies, indicating that the distribution of *D. rotundus* is primarily restricted by winter temperatures below 15 °C [29]. Previous studies have suggested that vampire bats are sensitive to temperature fluctuations, because effective thermoregulation is essential for their metabolic activity. Decreases and increases in temperature can directly affect bat activity, influencing the metabolic rates and activity patterns [38,57]. In warm-blooded animals, maintaining a constant body temperature is vital for proper physiological functioning. Changes in thermal conditions may necessitate behavioral adjustments and migrations in search of regions with milder temperatures [58,59].

Our results also align with those of studies that have analyzed the impact of climate change on bovine rabies cases. Researchers have reported that temperature has a positive effect on viral transmission and mortality associated with bovine rabies, whereas precipitation can have a negative effect on the frequency of bovine rabies outbreaks [30]. Studies indicate that this negative association with precipitation can be attributed to the effects of rain on bat foraging, as well as the difficulty of vampire bats in locating cattle when there is a reduction in the surface body temperature of livestock, affecting the thermoreceptors used by common vampire bats to locate their prey [50,60]. Furthermore, abundant rain after a heat wave can reduce the fertility of vampire bats because of decreased common activity among vampire bats and mating between different roosts [61]. These findings demonstrate that climate change can directly affect vampire bat species and, consequently, influence cases of rabies in humans either directly or indirectly.

Regarding the analysis of future scenarios, our model revealed a trend of change in the areas of disease occurrence. Areas located in the northern and northeastern regions continued to be consistently suitable under future climate conditions, while the southern region showed a reduction in suitability and the central–west region experienced an increase in climatic suitability. In other words, under the current trajectories of global warming, population growth, socioeconomic instability, and the loss of natural areas, there is a probability of climate change impacts, especially those related to temperature, in both the optimistic and pessimistic scenarios. When analyzing the economic development of some Brazilian regions, it is possible to identify activities related to agriculture, extensive livestock farming, mining, industrial activities, and agro-industries [62]. These activities have various environmental impacts that directly affect the biome and its ecosystem services, leading to modifications to the natural habitats of various animal species. These environmental changes also have significant implications for climatic conditions, human health, and the quality of life of the affected populations because they can influence the spread of various infectious diseases [63]. Moreover, predicting the most likely scenario is challenging because of the current noncompliance of most countries with carbon dioxide (CO_2_) reduction treaties, compromising the reliability of optimistic scenarios and potentially substituting them with pessimistic projections.

Studies conducted using spatial analyses to assess the environmental impact and climate change in relation to cases of bovine rabies in Uruguay have revealed that the outbreak of paralytic bovine rabies coincided geographically and temporally with the increased fragmentation of native grasslands to create mono-specific forests for timber and cellulose production [64,65]. Furthermore, these analyses have indicated that fragmentation, along with minimum winter temperatures, enhanced the connectivity between colonies of vampire bats, facilitating the sharing of feeding areas, and consequently increasing the spatial persistence of the rabies virus in these bat groups [65]. By comparing these findings with those of our modeling approach, we reaffirmed the influence of climate change, especially temperature, on the distribution and occurrence of new cases of rabies in both animals and humans. Modeling studies of the distribution of vampire bats in current and future scenarios have also demonstrated the existence of suitable habitats in much of North and Central America, as well as in the Brazilian plateau, for the occurrence of these bats [29]. Additionally, these studies identified temperature seasonality as a limiting factor for species expansion, which is consistent with the findings of the present study.

Rabies control is a challenging task due to the epidemiological profile of bats, requiring a comprehensive “One Health” approach, despite social vulnerabilities, and the lack of population awareness [66]. Several documented rabies outbreaks in Brazil and South America have occurred in riverine regions, indigenous communities, rural areas, and extractive reserves, particularly in low-income populations and children who come into contact with bats near sylvatic cycles [67,68,69]. These areas are characterized by a low population density, lack of access to electricity, vulnerable housing, and limited access to healthcare services [68]. In this context, there is an urgent need to adopt integrated approaches that consider human and wildlife aspects. Public awareness of the risks associated with contact with bats and prevention strategies are the key components of rabies control initiatives. Furthermore, socioeconomic disparities and unfavorable living conditions in certain areas exacerbate the challenges of disease outbreaks, underscoring the urgency for interventions targeting these vulnerable communities.

Our model has certain inherent limitations, such as the inability to discern whether the climate variables identified in this study indirectly influenced the occurrence of rabies by affecting wild animal populations or by triggering direct effects on viral particles or replication. Additionally, predictive modeling for future scenarios is not intended to accurately replicate real-life occurrences, but rather to indicate the potential areas where climate change may influence disease dynamics. In the context of rabies, in addition to climatic variables, economic factors, particularly those related to agricultural activities, play a significant role in disease dissemination [70]. An illustrative example of this interaction is the common practice of cattle migration, which is often observed in regions of the Amazon, in search of more favorable pastures. Previous studies have shown that the movement of cattle correlates with the movement of vampire bat populations. This joint migration of cattle and bats can lead to rabies outbreaks without strict dependence on climate change [13]. It is worth noting that the occurrence of rabies cases is often associated with the proximity of grazing areas, where susceptible animals may come into contact with vampire bats, establishing a critical link in rabies virus transmission [71]. Thus, a comprehensive understanding of rabies dynamics involves not only climatic factors, but also economic and livestock management considerations.

Nonetheless, the results obtained hold intrinsic value, as they can contribute to directing rabies prevention and control efforts. Additionally, they can be used for monitoring current disease occurrences, as well as for projections of future scenarios, integrated with other patterns, such as rabies transmission epidemiology, migratory flows of animal populations, and epidemiological outbreaks [72]. Predictive modeling in this approach unifies the elements of spatial and environmental epidemiology, providing a holistic understanding of disease transmission geography. This integration across disciplines is crucial for the surveillance and monitoring of infectious diseases, enabling the elucidation of the spatial and temporal contexts of past, present, and emerging diseases [73]. Moreover, it allows for an in-depth investigation of the role played by environmental changes in the climate and landscape in relation to disease transmission dynamics, enhancing the capacity to respond to and prevent these diseases [74,75].

## 5. Conclusions

This study provides significant preliminary insights into the influence of climatic conditions on various scenarios of global warming in human rabies. Our results demonstrate a remarkable performance in predicting climate suitability, with an ROC value of 0.90. Furthermore, the standout bioclimatic variables in the consensus model, namely, temperature seasonality and the average temperature of the coldest month, were aligned with the ecological requirements of *D. rotundus*, which is recognized as the primary host responsible for maintaining the sylvatic cycle of the rabies virus. Within the Brazilian context, we identified the areas with a high likelihood of climatic factors contributing to the emergence of rabies cases. Looking towards future scenarios, we observed regional variations in climate suitability. These fluctuations could imply both an increase and decrease in the cases, depending on the region under consideration. It is essential to emphasize that these conclusions play a fundamental role in monitoring and predicting human rabies cases, and provide insights to guide future epidemiological surveillance efforts. Furthermore, they serve as a starting point for subsequent investigations involving newly modeled projections and enrich our understanding of the mechanisms underlying rabies transmission dynamics in the context of climate change.

## Figures and Tables

**Figure 1 ijerph-21-00212-f001:**
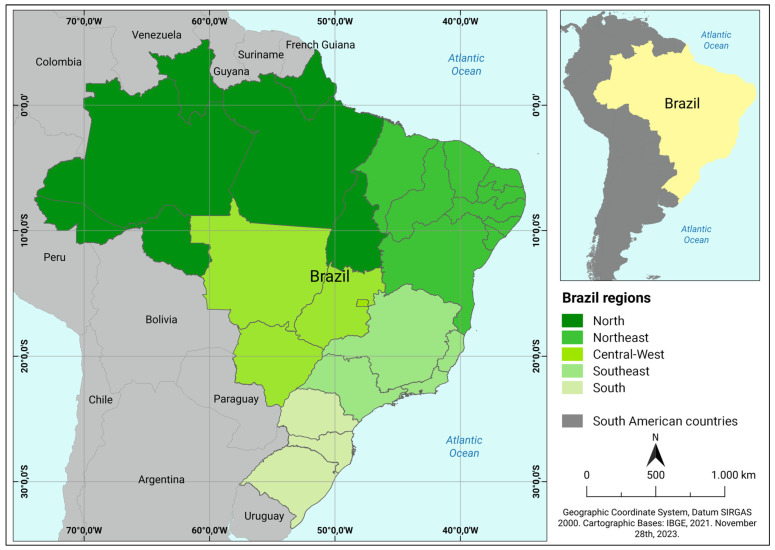
South American political map showing Brazil with international borders, neighboring countries, states, and regions.

**Figure 2 ijerph-21-00212-f002:**
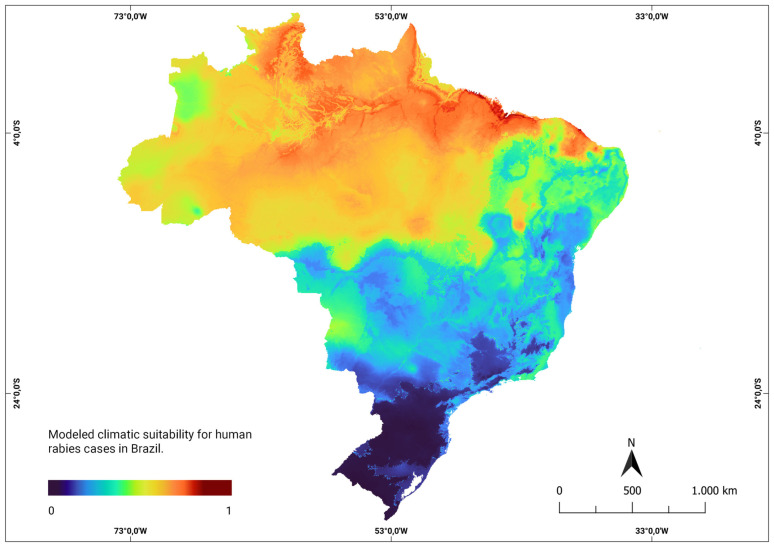
Modeled climatic suitability for human rabies cases in Brazil. Modeled climatic suitability (consensus model) for all seven algorithms under current climate conditions is presented. The data were provided by WorldClim [42]. For visualization, the maps were built using QGIS version 3.16 “https://qgis.org/pt_BR/site/ (accessed on 4 October 2022)”.

**Figure 3 ijerph-21-00212-f003:**
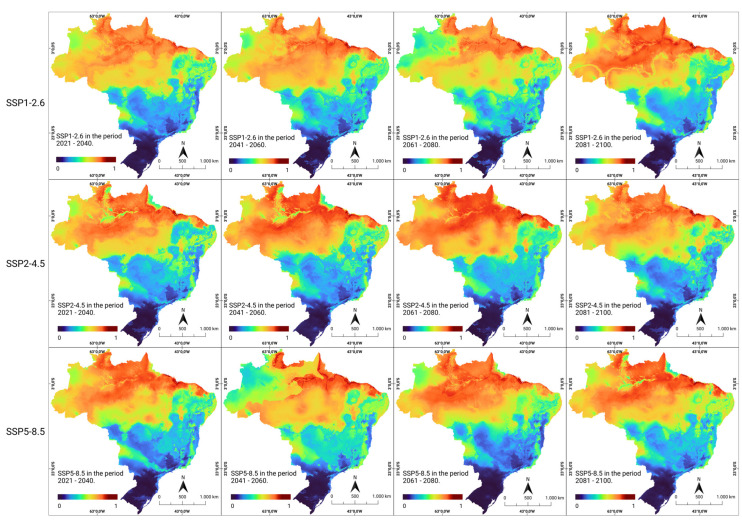
Future climate suitability for human rabies in Brazil. Modeled climate suitability (consensus model) for all seven algorithms under future climate conditions using three updated scenarios (SSP1-2.6, SSP2-4.5, and SSP5-8.5) during four CMIP6 periods in the period 2021–2100. The data were provided by WorldClim [42]. For visualization, the maps were built using QGIS version 3.16 “https://qgis.org/pt_BR/site/ (accessed on 4 October 2022)”.

**Figure 4 ijerph-21-00212-f004:**
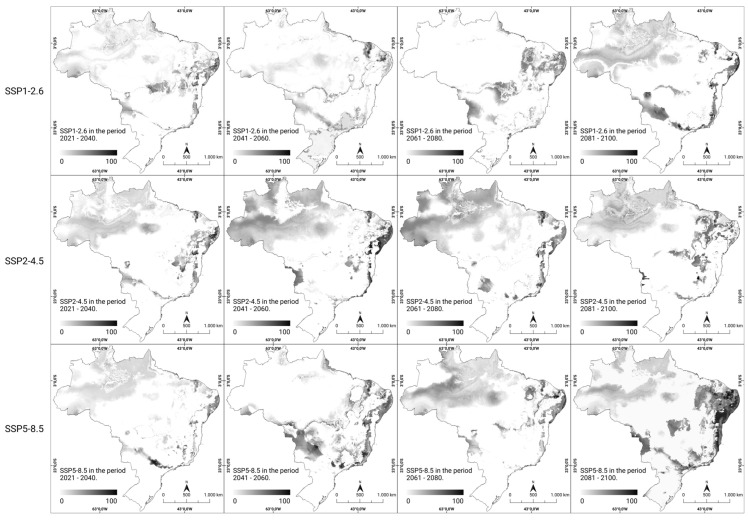
Percentage change in the predicted future climate suitability for human rabies in Brazil. The modeled climate suitability (consensus model) for all seven algorithms under future climate conditions using three updated scenarios (SSP1-2.6, SSP2-4.5, and SSP5-8.5) over four periods (2021–2100) from CMIP6 have been depicted. The data were provided by WorldClim [42]. For visualization, the maps were built using QGIS version 3.16 “https://qgis.org/pt_BR/site/ (accessed on 4 October 2022)”.

**Table 1 ijerph-21-00212-t001:** Values of the relative contribution of bioclimatic variables for human rabies modeling in the period from 2001 to 2023, Brazil.

Dependent Variables	Bioclimatic Variables	Statistical Algorithms						
		GLM	GAM	CTA	FDA	MARS	RF	MAXENT
Rabies Human	Temperature seasonality (BIO4)	0.514	0.447	0.721	0.427	0.258	0.706	0.266
	Minimum temperature of the coldest month (BIO6)	0.083	0.330	0.532	0.246	0.414	0.483	0.176
	Precipitation seasonality (BIO15)	0.019	0.216	0.245	0.141	0.186	0.578	0.122

## Data Availability

This data can be found here: JessicaMilenaMoura/Human-Rabies-Cases: Data and R code to support Neves et al., (2024). Modeling of Human Rabies Cases in Brazil in Different Future Global Warming Scenarios. International Journal of Environmental Research and Public Health. (github.com (accessed on 4 October 2022)).

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
