# Peer review of "Modeling of Human Rabies Cases in Brazil in Different Future Global Warming Scenarios"

_ijerph, 2024, doi:10.3390/ijerph21020212_

Round 1
Reviewer 1 Report
Comments and Suggestions for Authors
Dear Authors,
Your manuscript entitled "Modeling of human rabies cases in Brazil in different future global warming scenarios" has been reviewed,
This paper deserves attention since it highlights a very important topic related to the use of a new model to predict the possible human rabies cases in different global scenarios in Brazil. This prediction in based on available data related to the weather and the climate change in Brazil with regards to the numbers of human rabies cases detected in Brazil in the period between 2001 and 2023. This type of Study is very important a very helpful for the preparation against such kind of infectious disease due to the global warming.
This article is well Written in English Language, each section is well designed and well prepared,
Kindly find below a list of my comments, minor and major ones:
Minor Comments:
01- In the Keyword section, Line 36, You are kindly invited to add the term "Brazil" to these words.
02- In the Introduction section, Lines 41-46, in this part you talked about a fatality rate (approaching 100%), then you talked about 60.000 cases and 59.000 annual fatalities out of the (60.000 cases). The fatality rate is about 98.4% so it is better to put 98.4% instead of approaching 100%. It is more persuasive.
03- In the Materials and Methods section, Lines 162-166, When you talked about "These bioclimatic variables encompass... niche investigations, and research dedicated to climate change dynamics." This part needs references. I suggest the following ones:
Reference 01: Over half of known human pathogenic diseases can be aggravated by climate change
Reference 02: Prevalence, risk factors and seasonal variations of different Enteropathogens in Lebanese hospitalized children with acute gastroenteritis
Reference 03: Modeling the Climatic Suitability of COVID-19 Cases in Brazil
04- In the Materials and Methods section, Line 197, When you mentioned for the first time the term "greenhouse gas" you are kindly invited to add (GHG). Then in the rest of part you can use GHG instead of greenhouse gas. So you can remove greenhouse gas from the line 204.
05- In the Materials and Methods section, Lines 199-201, when you used for the first time the term SSP, you are invited to put it abbreviation before you use it and not after. So you can move the sentence in lines 201-202 to the line 199.
06- Concerning Table 1, Why it does not contain information regarding all the BIO? from BIO1 to BIO19?
07- In the Results section, Line 276, you mentioned that "the model highlighted areas represented in red on the map" can you please mentioned the information regarding the red color in the Materials and Methods section?
08- In the Discussion section, Lines 439-442, When you talked about environmental changes and transmission of diseases, I suggest to add the following articles as references for this point which empower your study:
Reference 04: Climate change and infectious disease: a review of evidence and research trends
Reference 05: Prevalence, antimicrobial resistance and risk factors for campylobacteriosis in Lebanon
09- References in the article (Last Pages) are not aligned and the text is not Justified.
Major Comments:
01- In the Materials and Methods section, you used the following terms SSP1-2.6, SSP2-4.5 and SSP3-8.5. What do you mean by 2.6, 4.5 and 8.5? This is not clear in the report.
02- In the Results section, Lines 268-269, you mentioned the follow "Among these variables, BIO4 showed the most significant relative contribution to the consensus models, as shown in Table 1." Based on which results you are saying this? In addition where are the results for all BIOs? (BIO1 to BIO19)??
03- In the Results section, Lines 272-274, These results are not shown!! Can you please explain the source of these information?
Best Regards,
Author Response
Comments from Reviewer 1
General Comments: Your manuscript entitled "Modeling of human rabies cases in Brazil in different future global warming scenarios" has been reviewed, this paper deserves attention since it highlights a very important topic related to the use of a new model to predict the possible human rabies cases in different global scenarios in Brazil. This prediction in based on available data related to the weather and the climate change in Brazil with regards to the numbers of human rabies cases detected in Brazil in the period between 2001 and 2023. This type of Study is very important a very helpful for the preparation against such kind of infectious disease due to the global warming.This article is well Written in English Language, each section is well designed and well prepared. Kindly find below a list of my comments, minor and major ones:
Specific comments and requested revisions:
Minor comments:
Comment 1: In the Keyword section, Line 36, You are kindly invited to add the term "Brazil" to these words.
Response 1: We agree with this and have incorporated your suggestion into the keyword section. (line 36, in red)
Comment 2: In the Introduction section, Lines 41-46, in this part you talked about a fatality rate (approaching 100%), then you talked about 60.000 cases and 59.000 annual fatalities out of the (60.000 cases). The fatality rate is about 98.4% so it is better to put 98.4% instead of approaching 100%. It is more persuasive.
Response 2: We agree with this and have incorporated your suggestion. (line 41, in red)
Comment 3: In the Materials and Methods section, Lines 162-166, When you talked about "These bioclimatic variables encompass... niche investigations, and research dedicated to climate change dynamics." This part needs references. I suggest the following ones:
Reference 01: Over half of known human pathogenic diseases can be aggravated by climate change
Reference 02: Prevalence, risk factors and seasonal variations of different Enteropathogens in Lebanese hospitalized children with acute gastroenteritis
Reference 03: Modeling the Climatic Suitability of COVID-19 Cases in Brazil
Response 3: Thank you for the suggestion. We agree, and we will incorporate the reference in line 163 of the Materials and Methods section. (in red)
Comment 4: In the Materials and Methods section, Line 197, When you mentioned for the first time the term "greenhouse gas" you are kindly invited to add (GHG). Then in the rest of part you can use GHG instead of greenhouse gas. So you can remove greenhouse gas from the line 204.
Response 4: Thank you for the suggestion. We agree, and we will add the term (GHG) in line 196 and remove the word "greenhouse gas" from line 202, adding (GHG) instead. The changes are highlighted in red.
Comment 5: In the Materials and Methods section, Lines 199-201, when you used for the first time the term SSP, you are invited to put it abbreviation before you use it and not after. So you can move the sentence in lines 201-202 to the line 199.
Response 5: We appreciate the suggestion. We agree with it and have amended the text in the Materials and Methods section, lines 198-201. (in red)
Comment 6: Concerning Table 1, Why it does not contain information regarding all the BIO? from BIO1 to BIO19?
Response 6: Thank you for your question. The biomod2 package in the R software provides functionalities to parameterize modeling options through "myBiomodOptions" and to run individual models using "myBiomodModelOut." By parameterizing and executing these individual models, we conduct performance tests on bioclimatic variables through available algorithms, evaluating those that exhibited superior performance based on metrics such as the ROC curve. Following this preliminary analysis, we selected the variables that demonstrated the best parameters for the seven chosen algorithms in the modeling process, subsequently employed in the final consensus model, as explained in lines 226-238. Therefore, Table 1 displays only the bioclimatic variables used in the final consensus model. The inclusion of values for all 19 bioclimatic variables would not be feasible, as we chose to select those that best fit the final consensus model based on statistical parameters.
Comment 7: In the Results section, Line 276, you mentioned that "the model highlighted areas represented in red on the map" can you please mentioned the information regarding the red color in the Materials and Methods section?
Response 7: Thank you for your question. The information about the colors is specified in the Materials and Methods section, specifically in lines 241-244 "... In this index, 0 represents areas with a low predicted probability of adaptation to climate conditions (indicated in blue), whereas 1 represents regions with a high probability of suitable climate conditions (indicated in red) for human rabies cases."
Comment 8: In the Discussion section, Lines 439-442, When you talked about environmental changes and transmission of diseases, I suggest to add the following articles as references for this point which empower your study:
Reference 04: Climate change and infectious disease: a review of evidence and research trends
Reference 05: Prevalence, antimicrobial resistance and risk factors for campylobacteriosis in Lebanon
Response 8: Thanks for the sugestion. We agree with this and incorporate it into the Discussion section. (line 446, in red)
Comment 9: References in the article (Last Pages) are not aligned and the text is not Justified.
Response 9: Thanks for the sugestion. We agree with this and incorporate it into the References. (in red)
Major Comments:
Comment 1: In the Materials and Methods section, you used the following terms SSP1-2.6, SSP2-4.5 and SSP3-8.5. What do you mean by 2.6, 4.5 and 8.5? This is not clear in the report.
Response 1: Thank you for your question. For the reader to gain a better understanding of the text, we have added a paragraph that provides further explanation of the scenarios used in the climate model in lines 249-255, highlighted in red.
Climate models are constantly updated as different modeling groups worldwide incorporate higher spatial resolution, new physical processes, and biogeochemical cycles. These modeling groups coordinate their updates according to the schedule of the Intergovernmental Panel on Climate Change (IPCC) assessment reports, releasing a set of model results. The sixth IPCC Assessment Report (AR6) introduced new state-of-the-art models called CMIP6. CMIP6 consists of "runs" from around 100 distinct climate models produced by 49 different modeling groups. Various scenarios have been produced for different ranges of carbon emission into the atmosphere. These updated additional scenarios are called SSP1-2.6, SSP2-4.5, SSP4-6.0, and SSP5-8.5. These scenarios include social variables such as population increase, economic factors, and bioecological aspects. The SSP1-2.6 scenario, titled "Sustainability," outlines a sustainable future characterized by low greenhouse gas emissions. This scenario assumes a rapid transition to cleaner energy sources, substantial investments in sustainability, and an increase in international cooperation. The SSP2-4.5 scenario, named "Middle of the Road," represents a trajectory of moderate emissions, reflecting a world where measures for emission reduction are implemented moderately, without extreme policy or technological changes. On the other hand, the SSP5-8.5 scenario, titled "Fossil-Driven Development," portrays a future where greenhouse gas emissions will continue to increase due to a persistent reliance on fossil fuels. This scenario implies more uneven economic development and less effective policies in emission reduction.
Comment 2: In the Results section, Lines 268-269, you mentioned the follow "Among these variables, BIO4 showed the most significant relative contribution to the consensus models, as shown in Table 1." Based on which results you are saying this? In addition where are the results for all BIOs? (BIO1 to BIO19)?
Response 2: Thank you for the question. As answered in comment 6 above, the biomod2 package in the R software provides functionalities to parameterize modeling options and to run individual models. By parameterizing and executing these individual models, we conduct performance tests on bioclimatic variables through available algorithms, evaluating those that exhibited superior performance based on metrics such as the ROC curve. Following this preliminary analysis, we selected the variables that demonstrated the best parameters for the seven chosen algorithms in the modeling process, subsequently employed in the final consensus model, as explained in lines 226-238. Therefore, Table 1 displays only the bioclimatic variables utilized in the final consensus model. Upon analyzing Table 1, it is observed that the variable BIO4 shows the highest contribution values for all chosen algorithms in the final consensus model. The inclusion of a table with the values of all 19 bioclimatic variables would not be feasible, as we chose to select those that best fit the final consensus model based on statistical parameters.
Comment 3: In the Results section, Lines 272-274, These results are not shown!! Can you please explain the source of these information?
Response 3: Thank you for the question. The main approach of the biomod2 package is to generate consensus maps, considering the overall mean derived from all models with ROC value > 0.80 and weighted by ROC. These consensus maps provide a robust estimate of species' climatic suitability, as opposed to individual models that may not accurately represent observed reality. The consensus strategy simplifies complex models into easily visualized maps, using a straightforward risk index ranging from zero to one, as explained in the Materials and Methods section (lines 232-238). This analysis accounts for the outcome shown in lines 272-274.

Reviewer 2 Report
Comments and Suggestions for Authors
The bioclimatic prediction study by Jessica Milena Moura Neves et al. potentially predicts the region-wise incidence of rabies in Brazil based on seven statistical modeling algorithms. Factors such as climate change due to global warming, loss of nature, population, etc were considered for the calculations. Change in local climate, and a couple other factors which may impact a certain bat population survival and growth were found to be most significant. This study is helpful as a caution.
Rabies has a high mortality rate. Although there are global vaccination programs for dogs and cats, rabies caused due to bats is still a major concern. Especially, vampire bat-borne rabies have been the major transmitter in recent years in Brazil (Horta el at, 2022, PLOS Neglected Tropical Diseases). This study discusses how climate change can change bat habitats, metabolism, geographical distribution, and other factors which may directly influence the future rabies scenario. This study is novel in the use of historical data from WorldClim website and using modeling softwares to predict essential parameters that may influence rabies occurrence in future, in Brazil. This study corelates climatic factors with human rabies cases for the years 2021-2100. Most studies based in Brazil till date have predicted cattle rabies (For eg., Braga et al, 2014, Preventative Veterinary Medicine; Santos et al, 2022, Epidemiology and Infection). Thus I believe this study raises a critical issue and is highly relevant to their field of infectious diseases. It fills the gap of predicting human rabies scenario till year 2100 in Brazil. I believe they answer their main questions in the article and have appropriate references in the introduction and discussion section. I am not an expert in modeling softwares hence I can not comment on the methodology section of the article.
I had a few minor suggestions:
It will be informative to have an additional figure dividing the regions based on current Brazil's ecosystems and bat habitats.
A table similar to table 1, but for all the bioclimatic variables can be added as a supplementary table.
Line 24: There should not be any space between "WorldClim".
Line 145: Are the data from " Information System for Notifiable Disease" published somewhere or is there a website? Please reference that.
Author Response
Comments from Reviewer 2
General Comments: The bioclimatic prediction study by Jessica Milena Moura Neves et al. potentially predicts the region-wise incidence of rabies in Brazil based on seven statistical modeling algorithms. Factors such as climate change due to global warming, loss of nature, population, etc were considered for the calculations. Change in local climate, and a couple other factors which may impact a certain bat population survival and growth were found to be most significant. This study is helpful as a caution. Rabies has a high mortality rate. Although there are global vaccination programs for dogs and cats, rabies caused due to bats is still a major concern. Especially, vampire bat-borne rabies have been the major transmitter in recent years in Brazil (Horta el at, 2022, PLOS Neglected Tropical Diseases). This study discusses how climate change can change bat habitats, metabolism, geographical distribution, and other factors which may directly influence the future rabies scenario. This study is novel in the use of historical data from WorldClim website and using modeling softwares to predict essential parameters that may influence rabies occurrence in future, in Brazil. This study corelates climatic factors with human rabies cases for the years 2021-2100. Most studies based in Brazil till date have predicted cattle rabies (For eg., Braga et al, 2014, Preventative Veterinary Medicine; Santos et al, 2022, Epidemiology and Infection). Thus I believe this study raises a critical issue and is highly relevant to their field of infectious diseases. It fills the gap of predicting human rabies scenario till year 2100 in Brazil. I believe they answer their main questions in the article and have appropriate references in the introduction and discussion section. I am not an expert in modeling softwares hence I can not comment on the methodology section of the article.
I had a few minor suggestions:
Comment 1: It will be informative to have an additional figure dividing the regions based on current Brazil's ecosystems and bat habitats.
Response 1: Thank you for the suggestion. We agree and have added a figure in Appendix 1 that depicts the current Brazilian biomes and the distribution of bat species in Brazil that tested positive for rabies. This figure is referenced in the article as (Appendix 1) in line 150 in blue.
Comment 2: A table similar to table 1, but for all the bioclimatic variables can be added as a supplementary table.
Response 2: I appreciate your suggestion. The biomod2 package in the R software provides functionalities to parameterize modeling options through "myBiomodOptions" and to run individual models using "myBiomodModelOut." By parameterizing and executing these individual models, we conduct performance tests on bioclimatic variables through available algorithms, evaluating those that exhibited superior performance based on metrics such as the ROC curve. Following this preliminary analysis, we selected the variables that demonstrated the best parameters for the seven chosen algorithms in the modeling process, subsequently employed in the final consensus model, as explained in lines 226-238. Therefore, Table 1 displays only the bioclimatic variables used in the final consensus model. The inclusion of a table with the values of all 19 bioclimatic variables would not be feasible, as we chose to select those that best fit the final consensus model based on statistical parameters.
Comment 3: Line 24: There should not be any space between "WorldClim".
Response 3: Thank you for the suggestion. We agree and have incorporated it in line 24. (in blue)
Comment 4: Line 145: Are the data from " Information System for Notifiable Disease" published somewhere or is there a website? Please reference that.
Response 4: Yes, they are published on DATASUS, a Brazilian open-access website with notifications of various diseases. Thank you for your suggestion. We agree with this change and will add this reference in line 145 as requested. (in blue

Round 2
Reviewer 1 Report
Comments and Suggestions for Authors
Dear Authors,
Thank you for all the modifications you made,
The article is better for publication in its present form,
BR,